# Dynamical Transitions in a One-Dimensional Katz–Lebowitz–Spohn Model

**DOI:** 10.3390/e21111028

**Published:** 2019-10-23

**Authors:** Alessandro Pelizzola, Marco Pretti, Francesco Puccioni

**Affiliations:** 1Dipartimento Scienza Applicata e Tecnologia, Politecnico di Torino, Corso Duca degli Abruzzi 24, 10129 Torino, Italy; 2INFN, Sezione di Torino, via Pietro Giuria 1, 10125 Torino, Italy; 3Consiglio Nazionale delle Ricerche–Istituto dei Sistemi Complessi (CNR-ISC), Via dei Taurini 19, 00185 Roma, Italy

**Keywords:** driven diffusive systems, totally asymmetric simple exclusion process, dynamical transition

## Abstract

Dynamical transitions, already found in the high- and low-density phases of the Totally Asymmetric Simple Exclusion Process and a couple of its generalizations, are singularities in the rate of relaxation towards the Non-Equilibrium Stationary State (NESS), which do not correspond to any transition in the NESS itself. We investigate dynamical transitions in the one-dimensional Katz–Lebowitz–Spohn model, a further generalization of the Totally Asymmetric Simple Exclusion Process where the hopping rate depends on the occupation state of the 2 nodes adjacent to the nodes affected by the hop. Following previous work, we choose Glauber rates and bulk-adapted boundary conditions. In particular, we consider a value of the repulsion which parameterizes the Glauber rates such that the fundamental diagram of the model exhibits 2 maxima and a minimum, and the NESS phase diagram is especially rich. We provide evidence, based on pair approximation, domain wall theory and exact finite size results, that dynamical transitions also occur in the one-dimensional Katz–Lebowitz–Spohn model, and discuss 2 new phenomena which are peculiar to this model.

## 1. Introduction

The Totally Asymmetric Simple Exclusion Process (TASEP, see References [1,2] for recent reviews), originally inspired by biological traffic problems [3,4], has now gained a prominent role in non-equilibrium statistical physics, becoming a paradigmatic minimal model, with a status analogous to that of the Ising model in equilibrium statistical physics. It can be simply generalized by including additional processes and/or interactions, obtaining models with a small number of parameters that exhibit rich stationary state phase diagrams, as generalized Ising models do in the equilibrium case.

The TASEP is defined on a one-dimensional lattice, whose nodes can be empty or occupied by one particle. Particles can hop only in one direction and the physics depends strongly on the boundary conditions. The most interesting situation is obtained with open boundary conditions [5], when the lattice is connected to particle reservoirs at both ends. The non-equilibrium stationary state (NESS) of the model is then determined by the particle densities of the boundary reservoirs.

Exact results are available for the NESS phase diagram [6,7,8,9], which can be rationalized in terms of the so-called theory of boundary-induced phase transitions [10], based on certain maximum and minimum current principles. It was observed [11,12,13] that, in order for this theory to be applicable to models with certain additional interactions, a specific choice of the coupling between the system and the boundary reservoirs must be made.

More recently, on the basis of exact results [14,15,16] on relaxation towards the NESS, a dynamical transition has been identified in the TASEP, that is a singularity in the relaxation rate which does not correspond to any NESS phase transition. The existence of this dynamical transition has also been observed in numerical investigations based on the density matrix renormalization group [17], although a physical interpretation has long been lacking. In the last few years, two of us contributed to showing [18,19,20] that this dynamical transition can be at least qualitatively predicted by suitable mean-field-like approaches and is also exhibited by certain models (for which exact results are not available) which generalize the TASEP by including additional processes or interactions, namely the TASEP with Langmuir kinetics [21,22] and the Antal–Schütz (AS) model [11].

The picture which seems to emerge is that, in those NESS phases where the bulk density is determined by one of the boundary reservoirs—that is, the so-called low-density (LD) and high-density (HD) phases—two phases can be identified on the basis of the behaviour of the relaxation rate. In one phase, dubbed *slow* in Reference [19] and typically located near a coexistence line (where the rate vanishes) in the NESS phase diagram, the relaxation rate depends on both reservoir densities. In the other phase, dubbed *fast* in Reference [19] and located away from coexistence lines, the relaxation rate depends only on the bulk density, that is, on just one reservoir density. A natural interpretation is that in the fast phase the slowest relaxation mode is a bulk one, while in the slow phase it is determined by the boundary.

Here we address the question of the existence of a dynamical transition in a one-dimensional (1D) version [12,13] of the Katz–Lebowitz–Spohn (KLS) model [23]. This 1D KLS model is more general than the AS model, since here the hopping rate between nodes *i* and i+1 depends on the states of both nodes i−1 and i+2 (in the AS model there is no dependence on node i−1). It also exhibits a richer NESS phase diagram, due to the appearance, for certain values of the model parameters, of 2 maxima in the bulk current-density relation (the fundamental diagram, as it is called in the traffic literature). As a consequence, a richer pattern of dynamical transitions can be expected.

The NESS phase diagram of the 1D KLS model with Glauber hopping rates has been determined in References [12,13] using a Markov chain approach to kinetics (MCAK), kinetic Monte Carlo (KMC) and the theory of boundary-induced phase transitions. A rich phase diagram has been obtained, which depends strongly on the choice of boundary conditions. Among the possible choices, the so-called bulk-adapted boundary conditions, or bulk-adapted coupling between system and reservoirs, play a special role, making the theory of boundary-induced phase transitions applicable.

In the present work we look for dynamical transitions in the 1D KLS model with Glauber hopping rates and bulk-adapted boundary conditions, using three different methods: the pair approximation (PA), a modified domain wall theory (mDWT) and extrapolation of exact finite size results. The PA, that we have already used in our study of the dynamical transitions in the AS model [19], is slightly more general than the MCAK, and its use is justified by a property of the NESS of our model. Under particular conditions, which are verified in the bulk in the large size limit, the NESS probability distribution is the equilibrium Boltzmann distribution of a 1D Ising (or lattice gas) model with nearest-neighbour interactions [13]. This makes the PA (and also the MCAK) exact for the NESS, and hence a good candidate, though no longer exact, for the analysis of the relaxation towards the NESS. In the PA the probability distribution of the system configuration is assumed to factor, at all times, into a product of its marginals, the largest marginal involved being that of 2 nearest-neighbour nodes. In other words, the PA can be regarded as a truncation of the entropy cumulant expansion, where the entropy cumulants beyond nearest–neighbours are assumed to vanish. The mDWT is a simple heuristic approach introduced in Reference [16], which in the TASEP case reproduces by construction the exact result for the relaxation rate and which we found useful in our investigation of the AS model [19]. Finally, the extrapolation of exact finite size results was first used in Reference [24], with very accurate results which could have led to an early discovery of the dynamical transition. Again, we found this technique useful in our study of the AS model [19].

## 2. Model and Methods

The 1D KLS model is defined on a linear chain of *N* nodes, labelled by i=1,…,N. Each node can be empty or occupied by a single particle. We introduce occupation number variables $n_{i}^{t}$ taking value 0 (respectively 1) if node *i* is empty (resp. occupied) at time *t*. A particle can hop in one direction only (say rightward, or from node *i* to node i+1), provided the destination node is empty. The hopping rate from node *i* to node i+1 depends on the configuration of the nodes i−1 and i+2 and will be denoted by $\Gamma_{i,i+1}\left(n_{i-1},n_{i+2}\right)$. Following References [12,13] we shall use the Glauber rates
(1)$$\Gamma_{i,i+1}\left(n_{i-1},n_{i+2}\right)\equiv \Gamma \left(n_{i-1},n_{i+2}\right)=\frac{1}{1+\exp[\left(n_{i+2}-n_{i-1}\right)V]}\;.$$

It can be shown [13] that, with this choice of the hopping rates the NESS probability distribution in the bulk (e.g., in a model with periodic boundary conditions, with a given particle density which will be independent of time) is the equilibrium Boltzmann distribution of a 1D lattice gas model with repulsive (for V>0) nearest-neighbour interaction *V* and thermal energy $k_{B}T=\beta^{-1}=1$, that is with a dimensionless Hamiltonian
(2)$$\beta H=V\sum_{i}n_{i}n_{i+1}-\mu \sum_{i}n_{i}\;,$$
where μ is a chemical potential, conjugate to the particle density, and the time index in the occupation number variables has been dropped, since we are dealing with a steady state. A slightly more general result actually holds [13]: the NESS is described by the above Hamiltonian if the hopping rates satisfy
(3)$$\begin{array}{cc} \Gamma (0,1) & =e^{-V}\Gamma \left(1,0\right)\;, \end{array}$$
(4)$$\begin{array}{cc} \Gamma (0,1)+\Gamma (1,0) & =\Gamma (0,0)+\Gamma (1,1)\;. \end{array}$$
These conditions are clearly satisfied by the Glauber rates Equation (Equation 1), but other choices are possible, for instance the AS model, where $\Gamma \left(0,n_{i+2}\right)=\Gamma \left(1,n_{i+2}\right)$. It is important to stress here that the NESS distribution does not satisfy detailed balance with respect to the dynamics of the 1D KLS model, so the NESS is truly out of equilibrium.

The relationship between the bulk NESS and the 1D lattice gas has important consequences, since the equilibrium distribution of a 1D lattice gas (or Ising) model factors in a simple way into a product of local marginals, and the PA becomes exact [25]. In order to be more specific it is useful to introduce some notation for marginal distributions and expectation values. In the following we will denote the marginal distribution for a cluster of nodes starting at *i*, at time *t*, by
(5)$$P_{i}^{t}\left[klm\ldots \right]\equiv P\left\{n_{i}^{t}=k\;,n_{i+1}^{t}=l\;,n_{i+2}^{t}=m\;,\ldots \right\}\;.$$
We will also denote the local densities and correlations by
(6)$$\begin{array}{cc} \rho_{i}^{t} & \equiv P_{i}^{t}\left[1\right]=\langle n_{i}^{t}\rangle \;, \end{array}$$
(7)$$\begin{array}{cc} \phi_{i}^{t} & \equiv P_{i}^{t}\left[11\right]=\langle n_{i}^{t}n_{i+1}^{t}\rangle \;, \end{array}$$
respectively. In the NESS we will drop the time index (e.g., the local density will be $\rho_{i}$) and in a bulk NESS we will also drop the position index (e.g., the bulk density will be ρ). The local densities and correlations can be used to parameterize the 1-node and 2-node marginals as follows
(8)$$\begin{array}{cc} P_{i}^{t}\left[1\right] & =\rho_{i}^{t}\;, \end{array}$$
(9)$$\begin{array}{cc} P_{i}^{t}\left[0\right] & =1-\rho_{i}^{t}\;, \end{array}$$
(10)$$\begin{array}{cc} P_{i}^{t}\left[11\right] & =\phi_{i}^{t}\;, \end{array}$$
(11)$$\begin{array}{cc} P_{i}^{t}\left[10\right] & =\rho_{i}^{t}-\phi_{i}^{t}\;, \end{array}$$
(12)$$\begin{array}{cc} P_{i}^{t}\left[01\right] & =\rho_{i+1}^{t}-\phi_{i}^{t}\;, \end{array}$$
(13)$$\begin{array}{cc} P_{i}^{t}\left[00\right] & =1-\rho_{i}^{t}-\rho_{i+1}^{t}+\phi_{i}^{t}\;. \end{array}$$

The factorization property of the 1D lattice gas, and hence of a bulk NESS of our model, can now be expressed by writing that *k*-node marginals (k≥3) factor according to
(14)$$P_{i}\left[n_{i}n_{i+1}\ldots n_{i+k-1}\right]=\frac{\prod_{l=i}^{i+k-2}P_{l}\left[n_{l}n_{l+1}\right]}{\prod_{l=i+1}^{i+k-2}P_{l}\left[n_{l}\right]}\;,$$
that is into a product of 2-node and 1-node marginals (see e.g., Reference [25] for a derivation of this property for an equilibrium 1D model with nearest-neighbour interactions, which includes the Hamiltonian Equation (Equation 2)). The PA [18,19], as well as the MCAK [12,13] and related techniques [26,27,28,29,30], is based on the assumption that the factorization Equation (Equation 14) holds at any time, and this explains why these techniques can give exact results for a bulk NESS.

In terms of entropy, we can introduce 1-node and 2-node cluster entropies (in units such that $k_{B}=1$)
(15)$$\begin{array}{cc} S_{i} & =-\sum_{n_{i}=0,1}P_{i}\left[n_{i}\right]\ln P_{i}\left[n_{i}\right]\;, \end{array}$$
(16)$$\begin{array}{cc} S_{i,i+1} & =-\sum_{n_{i}=0,1}\sum_{n_{i+1}=0,1}P_{i}\left[n_{i},n_{i+1}\right]\ln P_{i}\left[n_{i},n_{i+1}\right]\;. \end{array}$$
Of course, cluster entropies can be defined analogously for clusters of any length. In particular, for a *k*-node cluster, Equation (Equation 14) implies that
(17)$$S_{i,\ldots ,i+k-1}=\sum_{l=i}^{i+k-2}S_{l,l+1}-\sum_{l=i+1}^{i+k-2}S_{l}=\sum_{l=i}^{i+k-1}S_{l}+\sum_{l=i}^{i+k-2}{\tilde{S}}_{l,l+1}\;,$$
where we have also introduced the 2-node entropy cumulants [25,31]
(18)$${\tilde{S}}_{l,l+1}=S_{l,l+1}-S_{l}-S_{l+1}\;.$$
A simple consequence of Equation (Equation 17) is that the 3-node cumulants [25,31]
(19)$${\tilde{S}}_{l,l+1,l+2}=S_{l,l+1,l+2}-S_{l,l+1}-S_{l+1,l+2}+S_{l+1}\;,$$
together with all the higher order ones, vanish in a bulk NESS of our model. Considering the PA (and other techniques sharing the same factoring assumption), we can say that it approximates the kinetics by neglecting 3-node (and higher order) entropy cumulants.

Using the factorization Equation (Equation 14) we can write kinetic equations for 1-node and 2-node marginals, that is for $\rho_{i}^{t}$ and $\phi_{i}^{t}$, at the PA level. Introducing the probability current
(20)$$J_{i}^{t}=\sum_{k,n=0,1}P_{i-1}^{t}\left[k10n\right]\;\Gamma \left(k,n\right)$$
and marginalizing the full master equation of the model, one obtains
(21)$$\begin{array}{cc} {\dot{\rho }}_{i}^{t} & =J_{i-1}^{t}-J_{i}^{t}=\sum_{k,n=0,1}P_{i-2}^{t}\left[k10n\right]\Gamma \left(k,n\right)-\sum_{k,n=0,1}P_{i-1}^{t}\left[k10n\right]\Gamma \left(k,n\right)\;, \end{array}$$
(22)$$\begin{array}{ccc} {\dot{\phi }}_{i}^{t}\; & = & \sum_{k=0,1}P_{i-2}^{t}\left[k101\right]\;\Gamma \left(k,1\right)-\sum_{n=0,1}P_{i}^{t}\left[110n\right]\;\Gamma \left(1,n\right)\;. \end{array}$$
The above equations, derived in the Appendix A, are exact but not closed in that they depend on 4-node marginals. The PA is indeed a possible closure scheme, using the factorization Equation (Equation 14) as an approximation:(23)$$P_{i}^{t}\left[k10n\right]≅\frac{P_{i}^{t}\left[k1\right]\;P_{i+1}^{t}\left[10\right]\;P_{i+2}^{t}\left[0n\right]}{P_{i+1}^{t}\left[1\right]\;P_{i+2}^{t}\left[0\right]}\;.$$

Before considering the kinetics according to the above equations, it is necessary to discuss bulk solutions for the NESS in some detail, and then specify the boundary conditions. In order to find a bulk NESS solution, that is a solution being invariant with respect to both time and position, we set $\rho_{i}^{t}=\rho$, $\phi_{i}^{t}=\phi$, ${\dot{\rho }}_{i}^{t}={\dot{\phi }}_{i}^{t}=0$ and drop all time and node indices in the marginals. From Equation (22) we obtain
(24)$$\frac{P[1100]}{P[0101]}=\frac{\Gamma (0,1)}{\Gamma (1,0)}\;,$$
which in the PA becomes
(25)$$\frac{P[11]\;P[00]}{{P[01]}^{2}}=\frac{\Gamma (0,1)}{\Gamma (1,0)}\;.$$

In order to reduce the above equation to its simplest possible form we introduce the *bulk correlator*
(26)$$\eta \equiv \frac{P[10]}{P[1]\;P[0]}\;.$$
We can then write the 1-node and 2-node marginals as
(27)$$\begin{array}{cc} P[0] & =1-\rho \;, \end{array}$$
(28)$$\begin{array}{cc} P[1] & =\rho \;, \end{array}$$
(29)$$\begin{array}{cc} P[10] & =P[01]=\eta \rho \left(1-\rho \right)\;, \end{array}$$
(30)$$\begin{array}{cc} P[00] & =(1-\eta \rho )\;\left(1-\rho \right)\;, \end{array}$$
(31)$$\begin{array}{cc} P[11] & =\left[1-\eta \left(1-\rho \right)\right]\;\rho \;. \end{array}$$
Equation (Equation 25) can then be rewritten as
(32)$$\frac{1}{\eta^{2}}-\frac{1}{\eta }+\rho \left(1-\rho \right)\left[1-\frac{\Gamma (0,1)}{\Gamma (1,0)}\right]=0\;,$$
which establishes the relationship between ρ and η characterizing the bulk steady state, and in particular it allows us to determine η as a function of ρ as
(33)$$\eta \left(\rho \right)=\frac{2}{1+\sqrt{1-4\rho \left(1-\rho \right)\;[1-\Gamma (0,1)/\Gamma (1,0)]}}\;.$$

We can now compute the bulk current from Equation (Equation 20). Using Equations (Equation 25)–(31) we obtain
(34)$$J\left(\rho \right)=\Gamma \left(1,0\right)\left\{\eta (\rho )\;\rho \left(1-\rho \right)+[1-\eta (\rho )]\;(a+b\rho )\right\}\;,$$
where we have also defined
(35)$$a\equiv \frac{\Gamma (1,0)-\Gamma (0,0)}{\Gamma (1,0)-\Gamma (0,1)}\;,\;b\equiv \frac{\Gamma (0,0)-\Gamma (1,1)}{\Gamma (1,0)-\Gamma (0,1)}\;.$$
With the Glauber rates Equation (Equation 1) we obtain a=1/2 and b=0 for the present model. Equation (Equation 34) can also be used for the AS model, in which case one obtains a=0 and b=1. For pure TASEP instead one has a=b=0 and η=1.

The bulk current-density relation (fundamental diagram) Equation (Equation 34) is illustrated in Figure 1 as a function of both the repulsion *V* and the density ρ. It must be stressed that this result has already been obtained in References [12,13] (using the MCAK), where the current was also shown to exhibit a single maximum at density 1/2 for $V<V_{*}=2\ln3$ and 2 maxima, symmetric with respect to a minimum at density 1/2, for $V>V_{*}$.

In the following we consider a system with open boundary conditions, coupled to 2 particle reservoirs, characterized by particle densities $\rho_{L}$ (the left reservoir) and $\rho_{R}$ (the right one). Specifying the coupling between the system and the reservoirs, that is the rates of particle injection from the left reservoir and extraction to the right reservoir, is a delicate issue, and several choices are possible. We will use the so-called bulk-adapted boundary conditions [11,12,13,26], which make the theory of boundary-induced phase transitions applicable. These conditions can be defined in such a way that they would yield bulk NESS solutions in semi-infinite systems. In order to fix ideas, consider the left boundary: the injection rate at node 1 will depend on the occupation of node 2. We would like to determine this injection rate in such a way that a semi-infinite system with nodes i=1,2,…, coupled to a left reservoir of density $\rho_{L}$, exhibits a bulk NESS of the type discussed above, with density $\rho_{L}$. This can be achieved if the left reservoir is equivalent to another semi-infinite system (with nodes at i=0,−1,…) whose NESS is a bulk one with density $\rho_{L}$. Within this framework, the time evolution of our system, at the PA level, is still described by Equations (Equation 21) (for i=1,…,N) and (22) (for i=1,…,N−1), provided we define the 4-node marginals $P_{i}^{t}\left[k10n\right]$ pertaining sites “outside the system” (namely, for i=−1,0,N−2,N−1) in a suitable way. In particular, we have to generalize Equation (Equation 23) (which holds for i=1,…,N−3) to the following one
(36)$$P_{i}^{t}\left[k10n\right]=\frac{{\tilde{P}}_{i}^{t}\left[k1\right]\;{\tilde{P}}_{i+1}^{t}\left[10\right]\;{\tilde{P}}_{i+2}^{t}\left[0n\right]}{{\tilde{P}}_{i+1}^{t}\left[1\right]\;{\tilde{P}}_{i+2}^{t}\left[0\right]}\;,\;i=-1,\ldots ,N-1,$$
where we have defined
(37)$${\tilde{P}}_{i}^{t}\left[mn\right]=\left\{\begin{array}{cc} P_{L}\left[mn\right]\;, & i=-1,0\\ P_{i}^{t}\left[mn\right]\;, & i=1,\ldots ,N-1\\ P_{R}\left[mn\right]\;, & i=N,N+1 \end{array}\right.\;,$$
(38)$${\tilde{P}}_{i}^{t}\left[n\right]=\left\{\begin{array}{cc} P_{L}\left[n\right]\;, & i=0,1\\ P_{i}^{t}\left[n\right]\;, & i=2,\ldots ,N-1\\ P_{R}\left[n\right]\;, & i=N,N+1 \end{array}\right.\;,$$
where in turn $P_{L}\left[\cdot \right]$ and $P_{R}\left[\cdot \right]$ denote the bulk probabilities (Equations (Equation 27)–(31), together with Equation (Equation 33)) evaluated for $\rho =\rho_{L}$ and $\rho =\rho_{R}$, respectively. From the above equations one can derive those reported in Reference [13] for the boundary rates, if Glauber rates are chosen according to Equation (Equation 1).

With the above choice for the coupling between system and reservoirs, the 1D KLS model is now fully specified, and we can discuss how our methods, namely the PA, the mDWT and the extrapolation of finite size results, can be applied to characterize the NESS and the relaxation towards it.

The PA is based on Equations (Equation 21) (for i=1,…,N), (22) (for i=1,…,N−1) and (Equation 36) (for i=−1,…,N−1), which can be rewritten as a whole by introducing the (2N−1)-component vector $x^{t}=\left(\rho_{1}^{t},\phi_{1}^{t},\ldots ,\rho_{N-1}^{t},\phi_{N-1}^{t},\rho_{N}^{t}\right)$, as
(39)$${\dot{x}}^{t}=f\left(x^{t}\right)\;.$$
The NESS $x=(\rho_{1},\phi_{1},\ldots ,\rho_{N-1},\phi_{N-1},\rho_{N})$ will be given by the condition f(x)=0, and relaxation near the NESS will be described by the relaxation matrix *M*, with elements
(40)$$M_{ab}=-{\left.\frac{\partial f_{a}}{\partial x_{b}^{t}}\right|}_{x^{t}=x}\;,\;a,b=1,\ldots ,2N-1\;.$$
In particular, the smallest eigenvalue $\lambda_{1}$ of *M* is the slowest relaxation rate, that is the inverse of the longest relaxation time, which, for our investigation of the dynamical transition, is the quantity we are mainly interested in. It is important to stress that the PA gives exact results for the bulk NESS solutions, but not for the relaxation rate $\lambda_{1}$. It is therefore important to obtain estimates of $\lambda_{1}$ from independent techniques. In this work we will use the mDWT and the extrapolation of exact finite size results, as we did in the case of the AS model [19].

The mDWT was proposed in Reference [16], on the basis of a comparison between the standard DWT result for the relaxation rate of pure TASEP and the exact one. In the DWT, the relaxation rate is given by
(41)$$\lambda_{1}={\left(\;\sqrt{D_{R}}-\sqrt{D_{L}}\;\right)}^{2}\;,$$
where $D_{L,R}=J\left(\rho_{L,R}\right)/\left|\rho_{R}-\rho_{L}\right|$. It turns out that the DWT result is exact in the slow phase of pure TASEP, and the dynamical transition corresponds to a maximum of the DWT rate. In the mDWT, which is exact by construction for pure TASEP, one takes the DWT result in the slow phase and the maximum rate in the fast phase. Based on our study of the AS model [19], we do not expect the mDWT to remain exact for models with interactions, but just to provide an independent qualitative confirmation of the PA results.

Finally, the third independent approach we will use to estimate the relaxation rate of our model will be the extrapolation of finite size results, along the lines of References [24,32], where very accurate results were obtained for pure TASEP. The smallest nonzero eigenvalue of the transition matrix of the full master equation is numerically computed for a set of small *N* values, and the N→∞ limit is then extrapolated using the Bulirsch–Stoer (BST) algorithm [33,34]. The parameter ω, characterizing the leading term in the expected size dependence, has been set at 2 (as in our study of the AS model [19]), based on the exactly known finite-size behaviour of the relaxation rate for pure TASEP [14,16], after verifying numerically that this value gives near-optimal results according to the criterion proposed in Reference [34].

## 3. Results

In the present section we compute the relaxation rate of the 1D KLS model with bulk-adapted boundary conditions, for $V=2V_{*}$ (that is, in the case of the dashed line in Figure 1), as a function of the reservoir densities. We choose this particular value because in this case the NESS phase diagram, reported for convenience in Figure 2, has already been determined in Reference [13] by means of extremal current principles which turn out to be exact in the case of bulk-adapted boundary conditions. In Reference [13], it has been shown indeed that, as far as the NESS is concerned, this case exhibits all relevant new features of the model.

This NESS phase diagram exhibits 7 different phases and a symmetry with respect to the diagonal $\rho_{L}+\rho_{R}=1$ due to the particle–hole symmetry of the model. In the portion of the phase diagram corresponding to each phase we report an identifier of the phase (I to VII) and the value of the bulk density. This is indeed a quantity which characterizes the NESS because, in the thermodynamical limit N→∞, the local density tends to the bulk density over the whole lattice, except in boundary regions whose size remains finite as N→∞. Phases I and VI are LD phases, whose bulk density $\rho_{L}$ is determined by the left reservoir. Symmetrically, phases III and V are HD phases, whose bulk density $\rho_{R}$ is determined by the right reservoir. In addition, there are the 2 maximal current phases II and VII, whose bulk densities $\rho_{M1}$ and $\rho_{M2}$ correspond to the maxima of the fundamental diagram along the dashed line in Figure 1, and the minimal current phase IV, whose bulk density 1/2 corresponds to a minimum in the fundamental diagram. Among the various NESS transitions separating these phases, it is important to observe here that LD and HD phases are always separated by discontinuous (in the bulk density) transitions, denoted by solid lines in Figure 2. Exactly at these transitions, the 2 phases coexist.

In the following we will compute the relaxation rate $\lambda_{1}$, using the PA, the mDWT and extrapolation of finite size exact results, in the HD phases. The corresponding results for the LD phases can be obtained by symmetry, and $\lambda_{1}$ vanishes at the discontinuous transitions between LD and HD phases. In the maximal and minimal current phases instead the relaxation is not exponential, so the relaxation rate is not defined. More precisely, the relaxation is power-law in these phases, and the relaxation rate vanishes upon approaching these phases from LD or HD phases.

We will be especially interested in singularities in the relaxation rate, representing dynamical transitions, which separate regions where $\lambda_{1}$ depends only on the bulk density (that is, on just one reservoir density) from regions where it depends on both reservoir densities. In order to exemplify the various possible transitions, we shall consider three cases in detail, corresponding to the thin solid lines in Figure 2, that is $\rho_{R}=0.75$ (HD phase V), 0.98 (HD phase V) and 0.45 (HD phase III).

Before turning to a detailed analysis, let us anticipate our main results. For $\rho_{R}=0.75$ a situation similar to pure TASEP is found, that is a slow phase close to the coexistence line, separated from a fast phase by a dynamical transition. For $\rho_{R}=0.98$, due to the proximity of the minimal current phase, new features appear, suggesting the possibility of a new dynamical phase and additional dynamical transitions. Finally, for $\rho_{R}=0.45$, we clearly find 2 slow phases close to the coexistence lines. According to the PA, these slow phases are separated by 2 dynamical transitions from a central phase which is a TASEP-like fast phase, although this picture is not fully confirmed by BST results, maybe due to small system sizes.

As previously mentioned, we start our analysis with the simplest case, that is $\rho_{R}=0.75$, in HD phase V. The relaxation rate $\lambda_{1}$ as a function of $\rho_{L}$ is reported in Figure 3. The PA results (black lines) clearly suggest that, in the thermodynamical limit N→∞, 2 phases can be distinguished. For small $\rho_{L}$ (slow phase), close to the (discontinuous) transition with the LD phase VI, $\lambda_{1}$ depends on $\rho_{L}$, starting from 0 at the transition and then increasing. Notice that in this phase the size dependence is very weak, in particular the asymptotic value is approached with an exponential decay in *N*. For large $\rho_{L}$ (fast phase), $\lambda_{1}$ becomes independent of $\rho_{L}$ (while the size dependence can be appreciated, the approach to the asymptotic value is $∼N^{-2}$) and a singularity, corresponding to a dynamical transition, can be clearly identified. This is the same behaviour observed in pure TASEP [16,17,18]. We can approximately locate the dynamical transition by extrapolating, using the BST algorithm with ω=2, the relaxation rate in the fast phase for $\rho_{L}=1$ and by looking for the point where the N=400 results reaches this value. The corresponding estimate of the transition point, computed for $\rho_{R}\in \left(\rho_{M2},1\right)$, will be reported in the dynamical phase diagram with a blue line.

In Figure 3 we also report the results of the DWT (dashed red line) and the mDWT (solid red line). As in pure TASEP, the DWT result for $\lambda_{1}$ starts correctly from 0 at the transition with the LD phase VI, then increases with $\rho_{L}$ until it reaches a maximum. In the pure TASEP case, the behaviour after the maximum was shown [16] to be incorrect, and the mDWT was constructed by replacing the right portion of the curve with a constant, equal to the maximum. Notice that the mDWT estimate is much smaller than the PA one (as already observed in the case of the AS model [19]), but the location of the dynamical transition (solid red line in Figure 6) is rather close to the PA one.

We also report the results of the BST extrapolation (grey dots) of finite size exact results. The relaxation rate is computed, in a numerically exact way, for N=4÷24. The BST algorithm with ω=2 is then applied to the subsequences N=4÷24 to N=17÷24. The results seem consistent with the qualitative behaviour found in the other techniques and suggest that the PA (respectively the mDWT) overestimate (resp. underestimate) the relaxation rate.

We now move on to the case $\rho_{R}=0.98$ (Figure 4), which is still in the HD phase V but, due to the proximity of the minimal current phase IV, exhibits a richer behaviour. Both the PA and DWT estimates behave in the same way as above for sufficiently small and sufficiently large $\rho_{L}$. There is however an intermediate region, close to the minimal current phase IV, where new features appear. The PA estimate becomes nearly constant, whereas the DWT one exhibits another maximum, followed by a minimum. One might be tempted to think that 2 more singularities, corresponding to additional dynamical transitions, appear, and the apparently unphysical non-monotonical behaviour of the DWT could be fixed in the mDWT by replacing the oscillating part of the curve with a constant equal to the minimum (solid red line in Figure 4). The mDWT estimates of these additional dynamical transitions are reported in Figure 6 with dashed lines, and the fact that they start from 3-phase points, like the transition discussed above (and like those found in pure TASEP and in the AS model), is encouraging. On the other hand, the PA rate shows a very weak size dependence here, as it usually happens in the slow phase where the rate depends on $\rho_{L}$, so the evidence is contrasting. The BST extrapolation of finite size results is again qualitatively consistent with the PA and mDWT, which in this case both overestimate the relaxation rate for most $\rho_{L}$ values. In the inset we can see that the mDWT seems very accurate (based on the comparison with the BST results) for $\rho_{L}<0.05$, and then deviates from the BST results. This may suggest that a dynamical transition occurs near $\rho_{L}=0.05$, giving some support to the scenario with additional dynamical transitions, though this issue is far from being settled.

Finally, we switch to HD phase III and we consider the case $\rho_{R}=0.45$ (Figure 5). Here the PA rates seem to suggest the existence of 2 dynamical transitions, separating a central fast phase from 2 slow phases, close to the LD phases I and VI. The 2 maxima exhibited by the DWT estimate are also consistent with this scenario. The approximate location of these dynamical transitions (blue lines in Figure 6) can be found in the PA following the criterion outlined above. It is instead more tricky to find a heuristic modification of the DWT, because the location and height of the 2 maxima depend in a non-trivial way on $\rho_{R}$. In particular, for $\rho_{R}<0.43$ the highest maximum becomes the right one. A criterion which leads to results qualitatively consistent with the PA would be to choose the constant value of the rate in the fast phase equal to the lowest maximum, but this would lead to an apparently unphysical singularity in the dynamical transition lines close to $\rho_{R}=0.43$, where the 2 maxima exchange their role. For this reason we do not report the transition lines estimated according to this criterion in Figure 6. The BST results seem to confirm the DWT ones close to the LD phases I and VI, but not in the central region, supporting the scenario with 2 slow phases separated by a central fast phase. Let us note indeed that in the central region the BST results are affected by a very high noise level (in some cases giving extrapolated results out of the figure range), which suggests they are not reliable at all and that longer sequences (by now beyond our current computational resources) would be needed to obtain convergence.

The full dynamical phase diagram, including the dynamical transitions in the LD phases, obtained by symmetry, is reported in Figure 6.

## 4. Discussion

Dynamical transitions are singularities in the relaxation (towards a NESS) rate, which do not correspond to any transition in the NESS itself. They have been already found in the TASEP [16] and, with approximate methods, in both the AS model [19] and the TASEP with Langmuir kinetics [20]. Dynamical transitions occur in those NESS phases where the bulk density equals the density of a boundary reservoir (HD and LD phases) and separate a fast phase, where the relaxation rate $\lambda_{1}$ depends only on the bulk density (suggesting that the slowest relaxation mode is a bulk one), from slow phases, where $\lambda_{1}$ depends on the density of both boundary reservoirs (suggesting that the slowest relaxation mode is boundary–dependent). The terms fast and slow were introduced in Reference [19] on the basis of the observation that, keeping fixed the bulk density (that is the density $\rho_{R}$ of the right reservoir in the HD phases) and varying the density of the other reservoir ($\rho_{L}$ in the HD phases), the relaxation rate takes its maximum value in the fast phase. In the NESS phase diagram the slow phases are typically located close to HD–LD coexistence (discontinuous transition) lines, where $\lambda_{1}$ vanishes. In a slow phase then the relaxation rate grows from 0 at the coexistence line to its maximum value, which depends on the bulk density, at the dynamical transition line.

We have investigated dynamical transitions in the 1D KLS model, which generalizes the TASEP by assuming that the hopping rate depends on the occupation state of the 2 nodes adjacent to the nodes affected by the hop. The 1D KLS model is more general than the AS model, where the dependence is only on the adjacent node in the direction of motion, and this dependence reflects in a richer fundamental diagram, which can exhibit 2 maxima and a minimum, and consequently in a richer NESS phase diagram. Following Reference [13] we have chosen Glauber rates and bulk-adapted boundary conditions. We have used the same techniques as in our previous work on the AS model [19], namely the PA, the DWT and its heuristic modification (mDWT), and the BST extrapolation of exact finite size results. The use of the PA is well justified, since with Glauber rates and bulk-adapted boundary conditions many PA results for the NESS are exact.

Our results, summarized in the dynamical phase diagram in Figure 6, seem to confirm the general picture outlined above, with slow phases close to coexistence lines, separated from fast phases by dynamical transitions. In addition, the 1D KLS model seems to exhibit certain qualitatively new phenomena. In the HD phase V we find evidence of additional phase transitions, possibly related to the appearance of a new dynamical phase close to the minimal current phase IV (by symmetry, this also applies to the LD phase I). This evidence is however only partial, and needs to be verified by additional investigations. Moreover, in the HD phase III, the nature of the central phase, a fast phase according to the PA, is not clearly confirmed by the BST results, and further investigations are also needed in this case (again by symmetry, this also applies to the LD phase VI). In both cases, further work may involve increasing the largest lattice sizes that we used in the exact finite size approach and/or moving in the parameter space of the model. In particular, varying the repulsion *V* in the range $V>V_{*}$ can be expected to shrink, or enlarge, the portions of the phase diagram occupied by the various phases, possibly making the analysis of the dynamical transition numerically simpler in certain cases. The results for $V\leq V_{*}$ can instead be expected to be qualitatively similar to pure TASEP (V=0), hence less interesting from the point of view of the present study.

Finally, it would also be interesting to check how the overall picture is affected by a change in the boundary conditions. The bulk-adapted boundary conditions that we have used here, following Reference [13], are not the only possibility. For instance, in Reference [13], a different choice was considered, named equilibrated-bath couplings, which leads to a significantly different NESS phase diagram. It is reasonable to expect that the dynamical transitions would also be strongly affected by such a change in the boundary conditions.

## Figures and Tables

**Figure 1 entropy-21-01028-f001:**
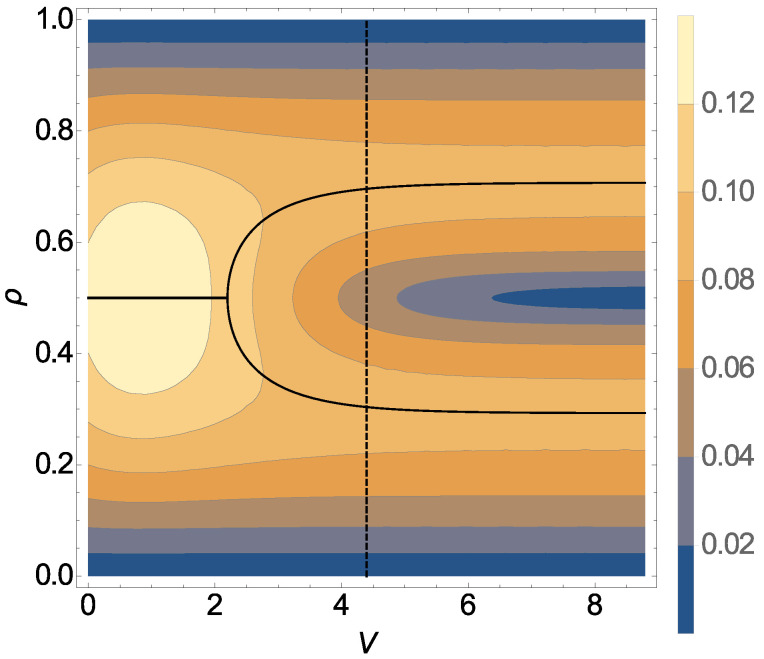
Contour plot of the bulk current as a function of repulsion *V* and density ρ. Maxima with respect to ρ are shown with full black lines. The vertical dashed line corresponds to $V=2V_{*}$.

**Figure 2 entropy-21-01028-f002:**
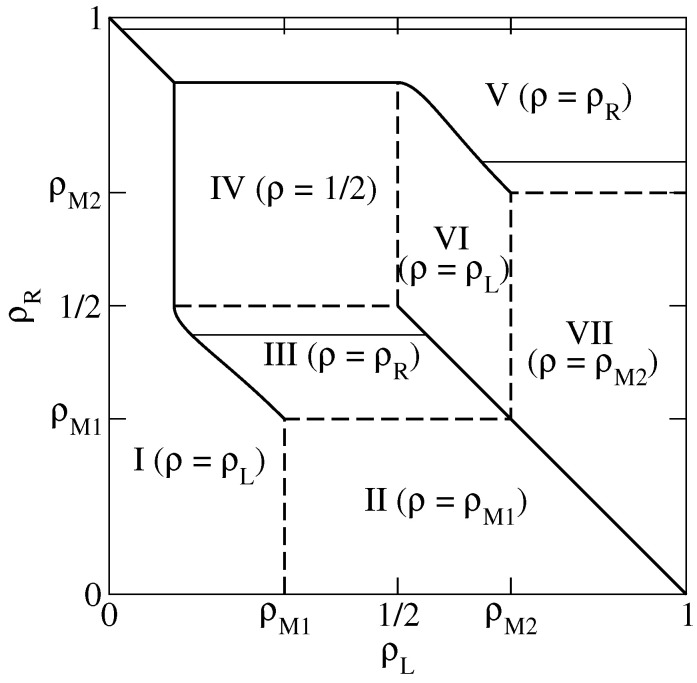
Non-equilibrium stationary state (NESS) phase diagram of the 1D Katz–Lebowitz–Spohn (KLS) model with $V=2V_{*}$. Solid (respectively dashed) thick lines denote discontinuous (resp. continuous) phase transitions. Solid thin lines are lines at constant $\rho_{R}=0.45,\;0.75$ and 0.98 in the high-density (HD) phases, where we perform our dynamical analysis.

**Figure 3 entropy-21-01028-f003:**
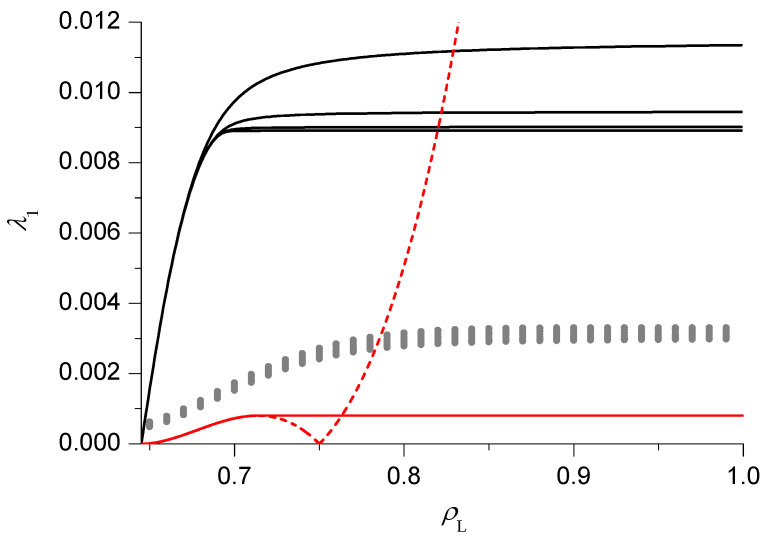
Relaxation rate $\lambda_{1}$ as a function of the left reservoir density $\rho_{L}$, for $V=2V_{*}$ and $\rho_{R}=0.75$, in the pair approximation (PA) (black lines, N=50,100,200,400 from top to bottom), domain wall theory (DWT) (dashed red line), modified DWT (mDWT) (solid red line) and extrapolation of finite size results (grey dots, see text for details).

**Figure 4 entropy-21-01028-f004:**
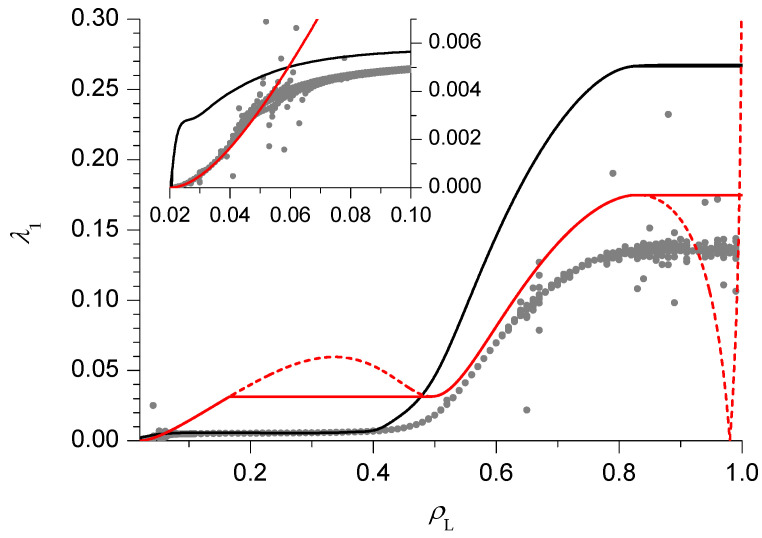
Same as Figure 3 for $\rho_{R}=0.98$.

**Figure 5 entropy-21-01028-f005:**
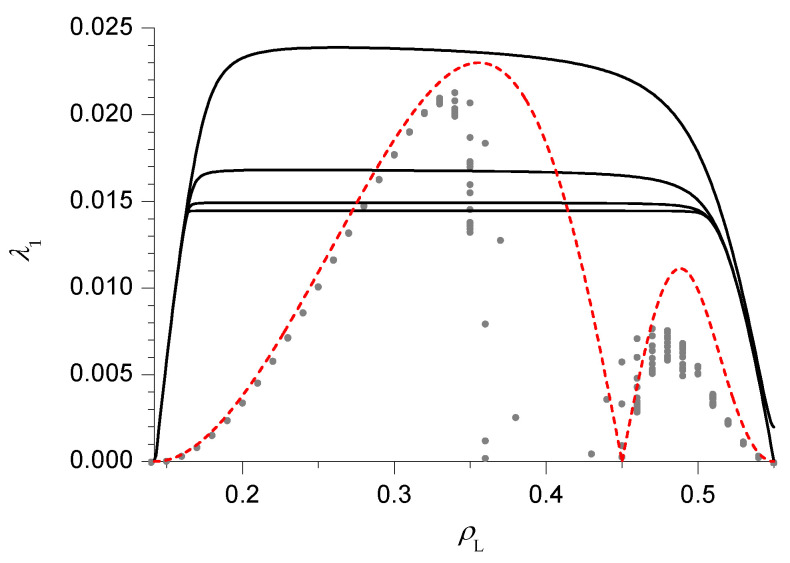
Same as Figure 3 for $\rho_{R}=0.45$.

**Figure 6 entropy-21-01028-f006:**
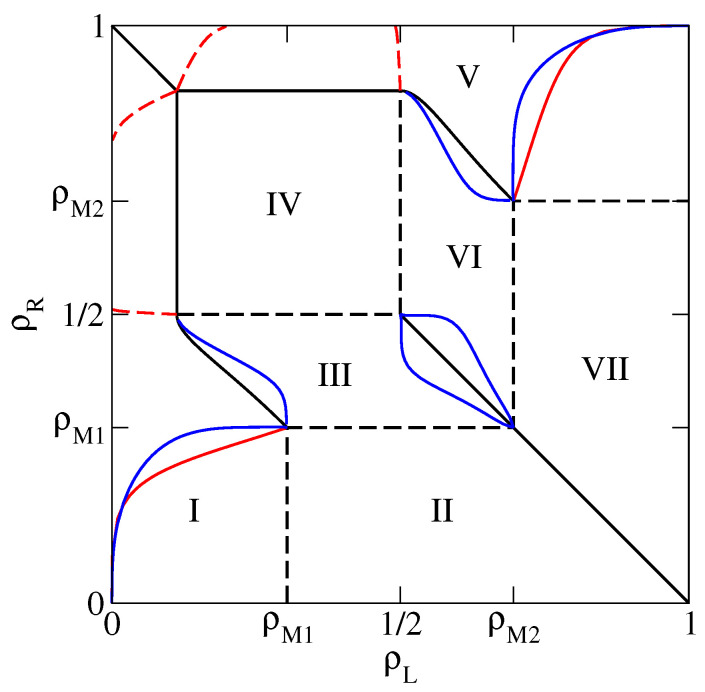
Same as Figure 2, with dynamical transition lines predicted by the PA (blue lines) and the mDWT (red lines). See text for the distinction between solid and dashed red lines.

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
