# Peer review of "Dynamical Transitions in a One-Dimensional Katz–Lebowitz–Spohn Model"

_entropy, 2019, doi:10.3390/e21111028_

Round 1
Reviewer 1 Report
This paper is devoted to explore the question of the existence of a dynamical transition in a one-dimensional version of the Katz-Leibowitz-Spohn model, in which the hopping rate (Glauber hopping rate) between nodes i and i+1 keeps some “memory” of the node i-1 and some “expectation” of node i+2.
One important strategy employed by the Authors is to treat the model in the pair approximation in the spirit of the Cluster Variation Method (as a truncation of the entropy cumulant expansion), which is a clever solution in view of the fact that this approximation is almost “exact” when applied to the analysis if the relaxation towards the Non-Equilibrium Stationary State. The other (auxiliary) strategies are the modified domain wall theory and the extrapolation of exact finite size results. The paper explores these techniques, having the pair approximation as the center of the analysis of the dynamical transitions.
I enjoyed reading this well written paper and think it is an excellent contribution to the investigations dealing with Totally Asymmetric Simple Exclusion Process and its (eventual) generalizations. It is important to add that the methodology employed by the Authors is original and innovative and is potentially applicable to a wide class of problems in similar contexts.
It is remarkable that their useful results, summarized in the dynamical phase diagram represented in Fig. 6, constitute a general picture of a whole class of phenomena, pointing also towards the existence of unexplored zones in this kind of phase diagrams.
I strongly recommend publication of this welcome and well-done contribution to the active field of research in the nonequilibrium statistical physics.
Author Response
We thank the reviewer for her/his report.
Reviewer 2 Report
The paper is very nice and very well written. The results are interesting
and sound. I suggest publication on Entropy, provided the following
minor comments are taken into account.
00. I am slightly confused by the use of the word equilibrium below line
85 on page 3. This evokes the validity of reversibility conditions and suggests
that (1) is reversible with respect to the Boltzmann distribution
with Hamiltonian (2). But this seems to be in contradiction with the
presence of current. The authors should clarify this point.
01. Explain what do you use to prove (14). In other words, what do you
mean by ``factorization property of the 1D lattice"? Do you use the
explicit expression (2) for the Hamiltonian?
02. Explain how (22) and (23) are derived. Add an appendix or
quote some proper reference.
03. Am I right if I say that (22) and (23) are exact? If so, please
stress it in the paper.
04. In equations (28)-(35) I would not use the symbol \bar\rho: it
generates confusion. I would write 1-\rho.
05. Explain the interest of the definition (27) of bulk correlator and
explain how it will be used in the sequel.
06. The authors should explain better the meaning of equation (37)
and should state clearly if equations (22) and (23) are exact for the
model with open boundaries.
07. Is the phase diagram in Figure 2 exact? If not, please explain
which approximation (or method) has been used to derive it.
08. In discussing the Figure 2, what do you mean exactly by "bulk
density"?
09. The discussion of the relaxation rate behavior is thorough
and clear. Nevertheless, I would appreciate a paragraph before
line 163 on page 9 in which the authors could summarize
their findings so that the reader can better be oriented in the
following discussion.
10. In Figure 6 I would remove the value or \rho. Symbols
I, ..., VII are sufficient.
11. Do the authors have any idea bout what would happen
for different values of V? say V<V_\star and V>V_\star.
I would be nice if the authors added a comment.
Author Response
We thank the referee for her/his report. Detailed replies follow.
00. We agree with the referee and, in order to avoid confusion, we have added an explanatory sentence after line 85.
01. The factorization property relies on the fact that interactions in Eq. (2) are among nearest--neighbours. We have added a sentence, referring to a review for a proof.
02. We have added an appendix to explain the derivation of Eqs. (22) and (23)
03. Correct. This is now discussed in detail in the appendix.
04. We have replaced \bar\rho with 1 - \rho.
05. The bulk correlator Eq. (27) has been chosen for simplicity: among the various quantities we have considered, it is the one which reduces Eq. (26) to its simplest form, Eq. (33).
06. This is also discussed in the appendix.
07. The phase diagram in Fig. 2 is indeed exact. It has been derived in [13] by means of extremal current principles which turn out to be exact in the case of bulk-adapted boundary conditions. We have added a sentence about this.
08. We have added a sentence which explains the meaning of "bulk density" in this context.
09. We have added a paragraph, before Fig. 3, which summarizes our main results for the relaxation rate.
10. We have removed rho values.
11. We have added a comment in the conclusions about possible results for different values of the repulsion V.